



# A new global high resolution wave model for the tropical ocean

Axelle GAFFET[1,2], Xavier BERTIN[2], Damien SOUS[3,4], Héloïse MICHAUD[5], Aron ROLAND[6], and
Emmanuel CORDIER[7]

[1]Créocéan, Zone Technocean - Chef de Baie, 10 Rue Charles Tellier, 17000 La Rochelle, France.
[2]UMR 7266 LIENSs, CNRS-La Rochelle Université, 2 Rue Olympe de Gouges, 17000 La Rochelle, France.
[3]Université de Pau et des Pays de l'Adour, E2S-UPPA, SIAME, 64600 Anglet, France.
[4]MIO, Université de Toulon, Bâtiment F, 83130 La Garde, France.
[5]Shom, 42 Avenue Gaspard Coriolis, BP 45017 - 31032 Toulouse CEDEX 5, France.
[6]BGS IT & E, Darmstadt, Hesse, Germany.
[7]Observatoire des Sciences de l'Univers de La Reunion (OSU-Réunion), UAR 3365, Université de La Réunion, CNRS, IRD,
Météo France, Saint-Denis, France.

**Correspondence:** Axelle GAFFET (axelle.gaffet@univ-lr.fr)

**Abstract.** Climate change is driving sea-level rise and potentially intensifying extreme events in the tropical belt, thereby increasing coastal hazards. In tropical islands, extreme sea levels and subsequent marine flooding can be triggered by cyclones but also distant-source swells. The knowledge of sea states in the tropical ocean is thus of key importance and their study is usually based on spectral wave models. However, existing global wave models typically employ regular grids with a coarse resolution, which fail to accurately represent volcanic archipelago, a problem usually circumvented by the use of obstruction grids but typically resulting in large negative biases. To overcome this problem, this study presents a new global wave model with a focus on distant-source swells, which received less attention than waves generated by cyclones. To accurately simulate sea-states in tropical areas, we have implemented the spectral wave model WAVEWATCH III© (WW3) over a global unstructured grid with a spatial resolution ranging from 50 km to 100 m. The model is forced by ERA5 wind fields, corrected for negative biases through a quantile-quantile approach based on satellite radiometer data. The wind input source terms adjusted accordingly and the explicit representation of tropical islands result in improved predictive skills in the tropical ocean. Moreover, this new simulation allows for the first time direct comparisons with *in-situ* data collected close to shore by water depths ranging from 30 m to 10 m.

## 1 Introduction

Over the last decades, coastal hazards have increased due to climate change, which causes sea-level rise as well as a possible intensification of extreme events in the tropical belt that can lead to more frequent marine floodings (Oppenheimer et al., 2019). Recent studies have even suggested that some low-lying islands and atolls could become uninhabitable by 2060-2090 due to annual flooding (Giardino et al., 2018). In addition, rising sea surface temperatures and ocean acidification exert strong pressure and locally degrade coral reefs, increasing the exposure of coastal islands to extreme events (Gattuso et al., 2014).

In tropical islands, extreme sea levels commonly result from storm surges, driven by the combination of atmospheric perturbations associated with tropical cyclones with a wave setup due to wave dissipation over reefs (Kennedy et al., 2012). Extreme




sea levels can also develop apart from cyclones due to distant-source swells (hereafter DSS) (Hoeke et al., 2013). DSS are generated by remote storms developing several thousand kilometers away from the tropical belt, with the resulting DSS then propagating toward tropical coasts (Munk et al., 1997; Delpey et al., 2010; Smithers and Hoeke, 2014). DSS are less studied

than cyclonic waves but their importance to coastal hazards was demonstrated at several tropical islands such as La Reunion Island (Lecacheux et al., 2012), French Polynesia (Canavesio, 2019; Andréfouët et al., 2023), Hawaii (Stopa et al., 2016), Marshall islands (Ford et al., 2018; Giardino et al., 2018) and British Virgin Islands (Cooper et al., 2013). It is worthwhile to highlight that, by contrast to cyclonic events associated to strong local winds and drop in atmospheric pressure, strong DSS events are able to impact and to damage the shore by physical processes solely driven by wave action (wave-setup, wave-driven

currents, direct wave impact or long wave generation).

To accurately represent the propagation of DSS over thousands of kilometers, from the swell source in high latitudes to the tropical oceans, global spectral models are nowadays the most efficient approach. Spectral models describe the space and time evolution of the wave energy spectrum, using a phase-averaged approach that typical employ a resolution of tens of kilometers in the deep ocean. However, the accurate representation of wave field around and within archipelagos made up of islands only

a few km-wide requires to reach a much finer resolution. To simulate the effect of small islands on the wave field while keeping computational times acceptable at global scale, an obstruction mask technique is classically used, where the percentage of land within each cell is used as a coefficient to calculate the attenuation of the wave action flux through the considered cell (Tolman, 2003; Mentaschi et al., 2018).

However, state-of-the-art hindcast usually exhibit substantial negative biases around archipelago, suggesting that this tech-

nique results in excessive wave energy loss (Rascle and Ardhuin, 2013; Dutheil et al., 2020). To overcome this issue, models using unstructured grids present an interesting potential since they allow small islands to be explicitly represented by locally refining the mesh. Unstructured grids have been first applied using explicit schemes, but the resulting high computational cost restricted this approach to small areas (Roland, 2008; Roland and Ardhuin, 2014). Through the adoption of implicit schemes which enable to overcome the CFL constraint and significantly reduce simulation duration (Abdolali et al., 2020), unstructured

grids slowly started to be used at a global scale (Brus et al., 2021; Mentaschi et al., 2023). Yet the spatial resolution used in these studies remains too coarse to allow for a direct validation around volcanic tropical islands because available *in-situ* data, coming from wave buoys or bottom-moored pressure sensors, are usually located very close to shore (i.e. less than one km) due to the steep seabed slope around islands.

Aiming to improve our capacity to accurately simulate sea states in tropical areas, we setup a new global spectral wave model

based on an unstructured grid with a resolution ranging from 50 km to 100 m. Such fine resolution is set to allow for direct comparisons with measurements available by water depth of 10 to 30 m, that is very close to shore. The subsequent sections of this paper are structured as follows. First, we describe the spectral wave model, its implementation and the observational data used for model validation at global and coastal scales. The section 3 highlights the model improvements through wind field corrections and the explicit representation of small islands together with global and local nearshore quantification of the model

performance. Finally, we discuss the added value and limitations of the present approach, including the remaining challenges associated with spectral wave modelling on unstructured grids in tropical area.



## 2 Material and methods

### 2.1 Global wave models

#### 2.1.1 Model description

Spectral wave models such as WAVEWATCH III© (WW3 Development Group, 2019) are typically used for large-scale applications, including operational or academic purposes. Spectral models are increasingly being applied to coastal areas, thanks to the development of unstructured grids versions, the better representation of coastal physics and the development of adaptive numerical schemes and integration strategy. In this work, we evaluate the performances of the model to simulate the sea states in the tropical ocean, using version 7.14 of WW3.

WW3 calculates the evolution of the wave spectrum by solving the Wave Action Equation (Komen et al., 1996). The source terms of this equation represent several key processes involved in wave transformation. In deep water, wave generation by wind ($S_{in}$) and wave dissipation by whitecapping ($S_{ds}$) are computed according to Ardhuin et al. (2010). Nonlinear wave interactions ($S_{nl}$) are modeled using the Discrete Interaction Approximation (DIA) of Hasselmann et al. (1985) and nonlinear triad interactions are modelled using the LTA model of Eldeberky (1996). The sea-ice interactions follow Liu and Mollo-

Christensen (1988); Liu et al. (1991); Ardhuin et al. (2015) for wave damping by ice ($S_{ice}$), and the approach of Moon et al. (2007) for scattering and dissipation by sea ice ($S_{is}$). In shallow water, two other source terms become important: wave dissipation by bottom friction ($S_{bot}$) which was developed during the SHOWEX experiments (Ardhuin et al., 2003) and the breaking-induced wave dissipation ($S_{db}$) which follows the formulation of Battjes and Janssen (1978).

For structured grids, the spatial propagation is done with the explicit third-order Ultimate Quickest scheme (Leonard, 1991).

This scheme is robust and stable but is limited by the CFL constraint, which discards the possibility to employ a spatial resolution fine-enough to capture nearshore wave transformations. Implicit schemes can be used to avoid prohibitive calculation costs at regional scale, while maintaining good scalability (e.g., Booij et al., 1999) on large core and mesh refinement. The WW3 implicit scheme, used in many studies and operational applications (e.g., Abdolali et al., 2020, 2021; Alves et al., 2022), computes a non-split solution of the wave action equation using a block Gauss-Seidel (Ferziger and Peric, 2002) solver for

source terms and advection, avoiding splitting errors in the usual fractional step method. Recently, the integration methods and the numerical limiter was reformulated following Hersbach and Janssen (1999). Under-relaxation (e.g., Moukalled et al., 2016) for the strong and nonlinear terms describing near-resonant triplet interactions and shallow water induced wave breaking was added. Additional improvements on parallelization were implemented (Roland, 2008; Abdolali et al., 2020). Here we use the domain decomposition methods based on ParMetis (Karypis, 2011), which is interfaced using PDLIB (Parallel Decomposition

Library).

#### 2.1.2 Model implementations

Two grids are implemented to assess the relevance of a new unstructured grid compared to classical structured grid.





The unstructured grid (hereafter UG) is created using SMS (Aquaveo, 2014). The mesh totals 296,199 nodes, with a resolution ranging from 50 km in the deep ocean to about 1 km around most islands. Around the selected validation sites (see Sect.

2.2.2), the mesh is further refined, reaching a resolution of 100 m at La Reunion island where field observations are available close to shore. All islands smaller than 10 km² were arbitrarily removed to limit CPU time. The periodic continuity between -180° and 180° is guaranteed through the modification of the grid connectivity table.

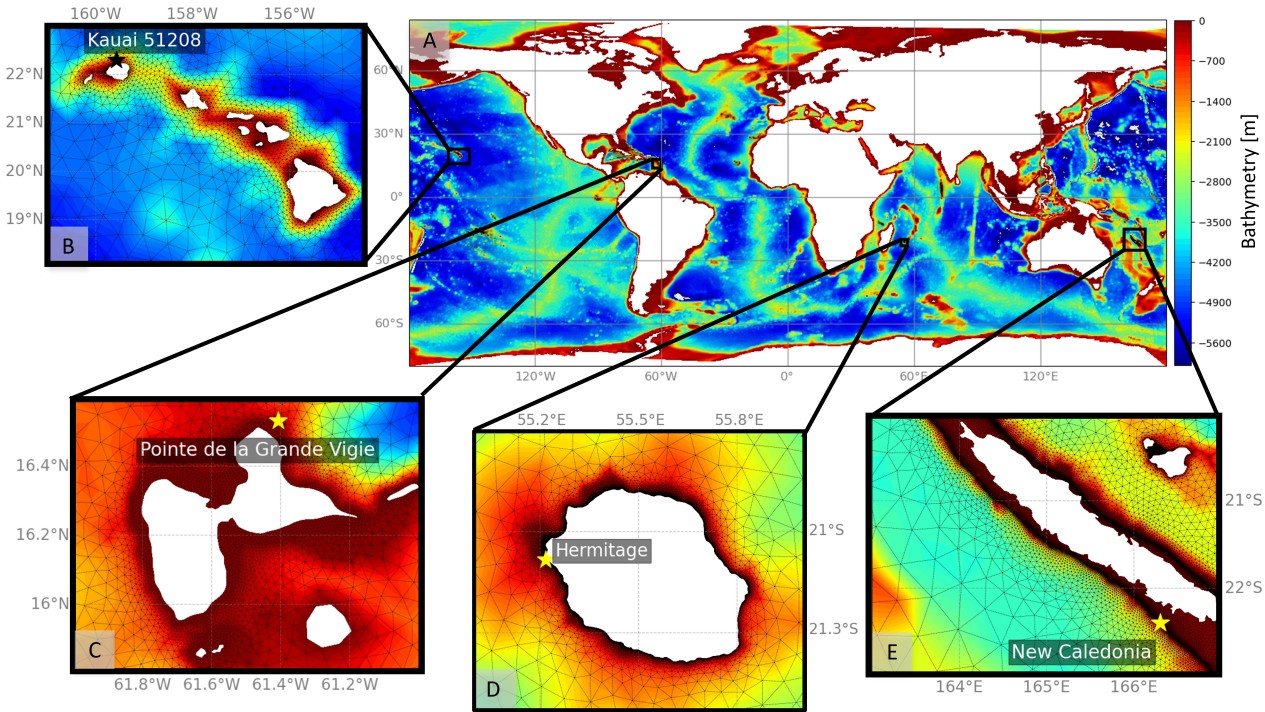

**Figure 1.** The global unstructured grid (A) with refinement at Hawaii (B), Guadeloupe (C), La Reunion (D) and New-Caledonia (E). The bathymetry is represented by the color legend. .

The structured grid (hereafter SG) has an uniform global resolution of 0.5° and islands smaller than the cell size are represented with an obstruction grid (Rascle and Ardhuin, 2013). The spectral grid consists of 24 directions and 36 frequencies

logarithmically spanning the range 0.035-1.01 Hz (meaning a frequency interval exponent of 1.1). For SG, time steps are set to: 1350 s for global, 450 s for geographic advection and 600 s for spectral advection. The minimum time step for source term integration is 35 s. For UG, the implicit scheme does not require any splitting and a time step of 800 s is set. The global bathymetry comes from the General Bathymetric Chart of the Oceans (GEBCO) dataset (Weatherall et al., 2015). For UG, a higher resolution bathymetry is used around French tropical islands (New-Caledonia, La Reunion Island, Mayotte, French

Polynesia, West Indies) originating from digital elevation models HOMONIM of 100 m resolution (Biscara and Maspataud, 2018).



Wind and ice forcing comes from the ERA5 reanalysis (Hersbach et al., 2020) with a 3-hourly time resolution and a spatial resolution of 0.5° globally. The ERA5 winds were preferred to the CFSR winds as they are more consistent over time (Liu et al., 2021). However, ERA5 winds are known to exhibit strong negative biases for the strongest winds (Pineau-Guillou et al., 2018; Campos et al., 2022). To solve this issue, Alday et al. (2021) proposed a correction of 5% of the wind speeds above 20 m/s and a new parameterisation of the source term in WW3 where the Betamax parameter, which controls the wave generation by the wind, is adjusted to 1.75. Our initial tests have shown that this approach reproduces well the most energetic sea states, but results in positive biases for calmer conditions. Given that ERA5 winds are unbiased for light to moderate winds, we developed an alternative strategy, where wind fields are corrected using a quantile/quantile approach, based on wind fields estimates from radiometers as described by Bentamy and Croize-Fillon (2012) and available since 1992. ERA5 wind fields were compared against radiometer data over different oceanic basins (Pacific, Indian, Atlantic) for selected 1-month periods (July 1996, May 2007, January 2014, respectively), encompassing major past swell events and enhanced very strong winds. In the Southern Pacific Ocean, a major storm in July 1996 produced one of the largest distant swells ever reported (Canavesio, 2019), with wind fields reaching 30 m/s (Fig. 2-C). In the Southern Indian Ocean, a strong storm in May 2007 produced winds over 30 m/s (Fig. 2-B), which drove a major distant swell (Lecacheux et al., 2012). Finally, in the NE Atlantic Ocean, winter 2014 exhibited an unprecedented succession of violent storms (Masselink et al., 2016), with several events driving winds over 35 m/s (Fig. 2-A). For these three periods and regions, the comparison confirms that ERA5 winds are unbiased for speed up to 15 m/s while for higher winds, the bias correction reaches 10% at 20 m/s and 15% at 25 m/s. Remarkably, this correction holds for all oceanic basins and time periods where the comparison is performed. Finally, the "Test471" parameterisation of Rascle and Ardhuin (2013) is used for the wind growth and whitecapping dissipation source terms, where the Betamax parameter has been adjusted to 1.43. The new parameterisation proposed in this study is hereafter called "ERA5_QC".

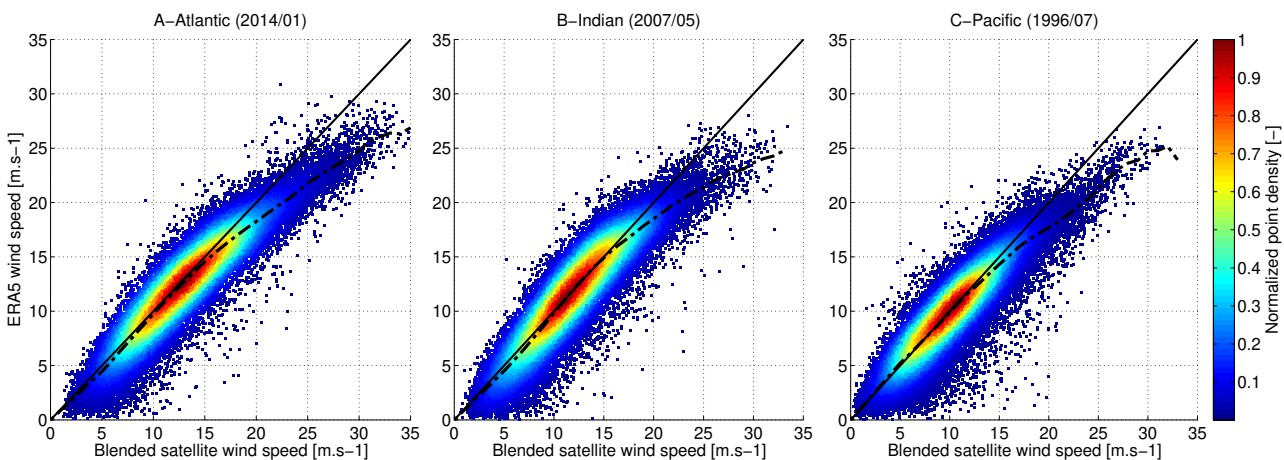

**Figure 2.** Comparisons between blended winds and ERA5 winds for A) The Atlantic ocean B) The Indian Ocean and C) The Pacific Ocean.

Other source terms are set to default settings.



## 2.2 Observational data

Altimetry data and *in-situ* measurements are used for global and coastal validation, respectively.

### 2.2.1 Altimetry data for global validation

The model evaluation in the deep ocean is performed against the European Space Agency Climate Change Initiative (ESACCI) altimetry data (Dodet et al., 2020; Schlembach et al., 2021). The ESACCI dataset v3 used here was retrieved from multiple satellite missions spanning from 2002 to 2022. For the present model validation, an entire year (2007) is used to cover a wide range of sea states. The simulated significant wave height $H_{m0}$ over the full spectral range is interpolated in time and space to match the satellite "denoised" significant wave height at approximately 6 km spatial resolution (Quilfen and Chapron, 2021; Schlembach et al., 2021). Both satellite data and interpolated WW3 outputs are averaged over $0.5°$ grid cells, which enables to calculate stable statistical values for the collocated satellite/model data. All satellite values out of the model time and space ranges were skipped. Coastal values in a 50 km range from the shoreline were flagged out due to possible coastline interference with the signal and the lack of model resolution along the coast.

### 2.2.2 Coastal and nearshore data

For each coastal/nearshore site, the model validation is performed over a four-month period, selected to represent a wide variability of wave conditions. The WW3 wave bulk parameters ($H_{m0}$ and $T_{m02}$) are calculated from the modeled spectrum using specific frequency cut-off, depending on the water depth or the device employed (see Table 1). For bottom pressure data, the nonlinear fully dispersive reconstruction described by Martins et al. (2021) is employed. Wave bulk parameters are then computed using classical spectral analysis.

**Table 1.** Description of coastal and nearshore data used for model validation at the four study sites

| Name | Lat | Lon | Depth | Data type | High frequency cut-off |
|---|---|---|---|---|---|
| Hawaii/51208 | 22.285 N | 159.574 W | 200 m | Wave Buoy (Datawell WR) | 0.62 Hz |
| Guadeloupe/97103 | 16.536 N | 61.407 W | 90 m | Wave Buoy (Datawell WR) | 0.62 Hz |
| La Reunion | 21.083 S | 55.217 E | 12 m | Pressure Sensor (RBR Duo) | 0.25 Hz |
| New-Caledonia/1402 | 22.397 S | 166.303 E | 11 m | Pressure Sensor (Seabird SB26Plus) | 0.25 Hz |

For Hawaii (Pacific Ocean), the validation data are provided by the 51208 wave buoy from the National Data Buoy Center (NDBC) and the period from August to December 2022 was chosen due to the occurrence of high energetic events. The data are owned and maintained by Pacific Islands Ocean Observing System (PacIOOS) and provided by the Scripps Institution of Oceanography. The buoy is located above Hanalei Bay, to the North of Kauai by 200 m water depth (see Table 1). For this buoy, only $H_{m0}$ and $T_p$ time series were available.

For Guadeloupe (Atlantic Ocean), the validation data are collected by the 97103 Candhis wave buoy from CEREMA (*Centre d'études et d'expertise sur les risques, l'environnement, la mobilité et l'aménagement*) and Météo-France. The period from





March to July 2008 was chosen for validation due to the occurence of an extrem swell event (Lefèvre, 2009). The buoy is located in the French West Indies, above the Pointe de la Grande Vigie, Northeast of Guadeloupe by 90 m water depth (see Table 1).

For La Reunion Island (Indian Ocean), the pressure sensor data are collected continuously in the framework of the SNO *Service National d'Observation* ReefTEMPS (Cordier et al., 2024), operated by the OSU-Reunion. The period from April to August 2022, during which a strong swell event occurred, was chosen for validation. The pressure sensor is located in the reef slope in front of the Hermitage Beach, bottom-moored by a mean water depth of 12 m.

For New-Caledonia (Pacific Ocean), pressure sensor data were collected from October 2019 to November 2020 during the GEOCEAN-NC 2019 field campaign (Chupin et al., 2023). The pressure sensor was moored in the reef slope to the Southwest of the main island by a mean water depth of 11 m (see Fig. 1). The period from March to July 2020 was chosen for validation as it encompasses several energetic events.

Lastly, as the 3-hour resolution of the wind forcing does not allow to capture the high frequency wave variability, observational data were low-pass filtered with a 3 h window.

## 3  Results

### 3.1  Global validation in deep water

The model validation in deep water is first performed on structured grids to evaluate the effect of wind field correction. The evaluation of the spatial discretisation is then carried out comparing SG and UG approaches, both including the wind correction.

### 3.1.1  Impact of wind field correction

Initial results with the parameterisation T475 of Alday et al. (2021) on SG show a global mean NRMSE of 14.1% and a global positive bias of 3.36% (see Fig. 3-A). Around island archipelagos (e.g French Polynesia, Indonesia, West Indies, Maldives), higher errors occur locally exceeding 30%. These larger errors are mostly associated to strong negative biases and are presumably linked to the obstruction grid. The implementation of the wind correction leads to better results, especially in the Atlantic, Indian and Pacific oceans, away from island areas, with a global NRMSE of 12.9% (Fig. 3-B). Strong positive biases are locally lowered which results in a global mean normalized bias of -4.4%.





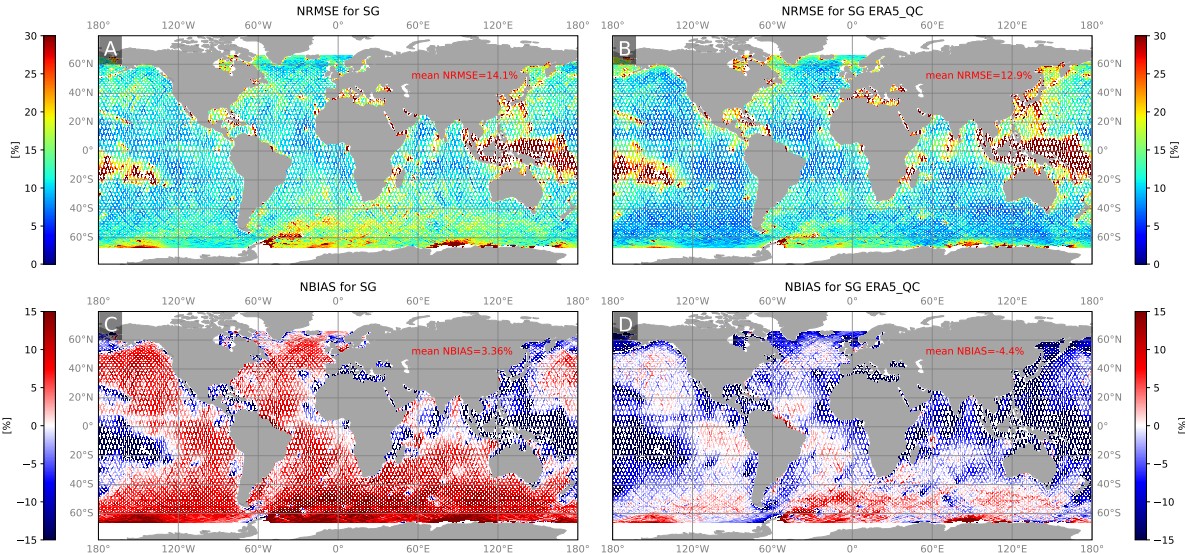

**Figure 3.** Normalized RMS error (A) and normalized bias (C) between $H_{m0}$ deduced from altimetry and simulated with T475 parameterisation and normalized RMS error (B) and normalized bias (D) with `ERA5_QC` parameterisation.

### 3.1.2  Impact of spatial discretisation: SG versus UG

Figure 4 displays the comparison between SG and UG configurations, both employing `ERA5_QC`. The unstructured grid leads to a lower global NRMSE of 11.6% (see Fig. 4-B). Much stronger local improvements are observed around archipelagos all around the globe. In SG, tropical archipelagos are generally associated with large NMRSE and strong negative bias. In UG, NRMSE is drastically reduced while the bias shifts to weakly positive. Restricting the NRMSE computation to the tropical band, between 23.27°N and 23.27°S, the error drops from 15.3% to 11% with the UG. The global mean normalized bias is considerably reduced to 1.5% with `ERA5_QC` parameterisation.

The results obtained globally for the UG `ERA5_QC` configuration will be further analyzed and discussed in Sect.4, where we investigate the implications of the remaining biases and their potential sources, particularly focusing on regions such as Antarctica, semi-enclosed seas, coastlines, the Mozambican channel, and the inner seas of Indonesia where higher errors persist.





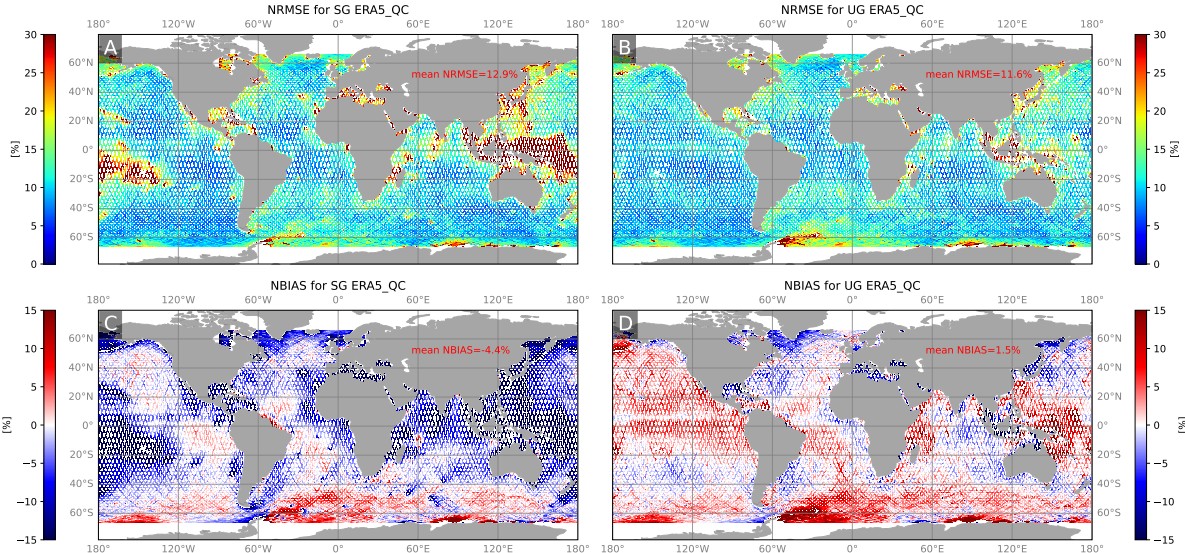

**Figure 4.** Normalized RMS error (A) and bias (C) between $H_{m0}$ deduced from altimetry and simulated with the UG `ERA5_QC` configuration and normalized RMS error (B) and normalized bias (D) with the SG `ERA5_QC` parameterisation.

The distribution of the bias as a function of $H_{m0}$ is shown on Fig. 5 for the three model configurations. With SG, the `ERA5_QC` parameterisation results in lower bias compared to T475. Below 1 m, $H_{m0}$ the . For the UG `ERA5_QC`, the bias is positive for $H_{m0}$ between 1 and 2.5 m and the bias slowly decreases after and is negative for $H_{m0}$ values ranging from 2.5 m to 12 m. The bias reaches its lowest value at -0.4 m for $H_{m0}$ of 11 m.

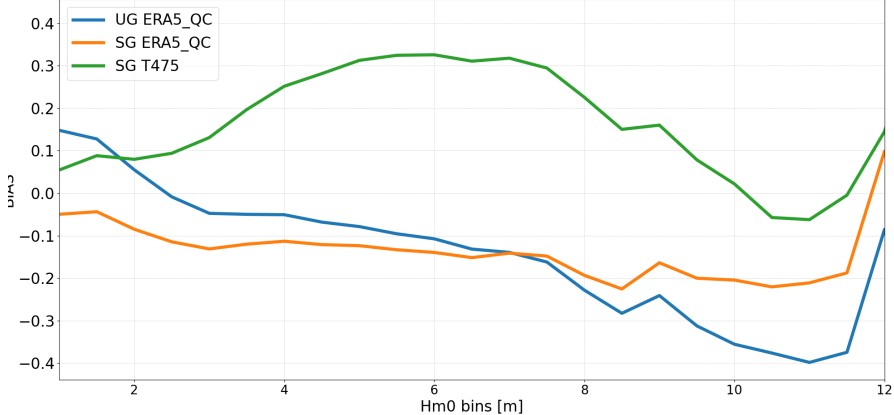

**Figure 5.** Distribution of the bias as a function of $H_{m0}$ for each simulation (SG T475, SG `ERA5_QC` and UG `ERA5_QC`) for the year 2007 at global scale.





## 3.2 Example of coastal validation

Long-term *in-situ* wave data are scarce in the nearshore area of tropical island bordered by steep slopes. The coastal validation of the model is carried out at four tropical islands: La Reunion in the Indian Ocean, Guadeloupe in the Atlantic Ocean, New-

Caledonia and Kauai (Hawaii archipelago) in the Pacific Ocean. These islands are representative of steep-slope tropical islands exposed to multi modal sea states. For each island, *in-situ* measurements are available by water depth ranging from 11 m to 200 m for periods over one year, allowing for the first time to evaluate the performance of a global wave model very close to shore.

### 3.2.1 La Reunion

Figure 6 displays the comparison between model and bulk parameters computed from pressure sensor data at La Reunion (12 m water depth) for $H_{m0}$, $T_{m02}$ and $T_p$. Overall, the model is able to capture the variability of the sea state over the selected time period, including the extreme DSS of June 29th, 2022 where $H_{m0}$ reached 7 m with $T_p$ exceeding 21 s. The NRMSE for $H_{m0}$ of 19.82% is similar to the NRMSE obtained in the deep ocean around La Reunion (see Fig. 4). The consistency between the results obtained globally and the results obtained in shallow water at the Hermitage demonstrates the ability of the model

to represent the wave transformation from deep ocean to shallow area. $H_{m0}$ tends to be slightly overestimated, which also matches the deep water bias in this region (Fig. 4). Wave periods are well represented with NRMSE of 9.31% and 8.29% for $T_{m02}$ and $T_p$, respectively.





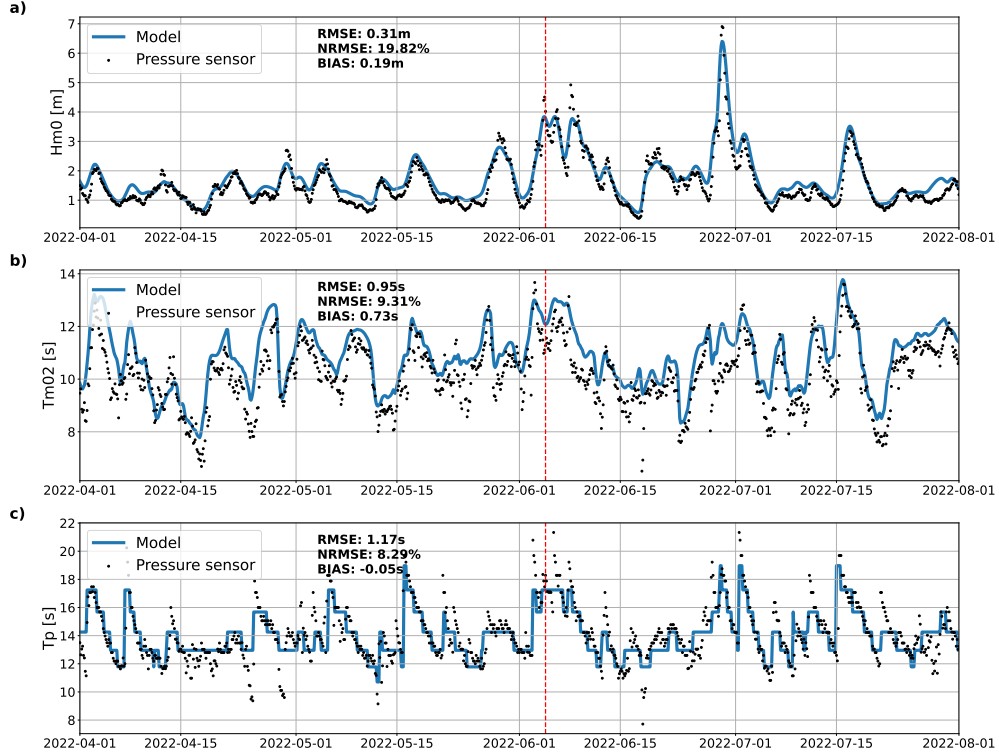

**Figure 6.** Wave bulk parameters derived from the pressure sensor against the model for April to August 2022 at Hermitage (La Reunion) for a) $H_{m0}$ b) $T_{m02}$ and c) $T_p$. The red line corresponds to a particular time where the PSD will be compared in the discussion.

### 3.2.2 Hawaii

Figure 7 shows the comparison between model and wave buoy data at Hawaii (200 m water depth) for $H_{m0}$ and $T_p$. The general temporal evolution of the wave bulk parameters is correctly represented, with $H_{m0}$ and $T_p$ NRMSE of 15.82% and 12.99%, respectively. However, the peaks in $H_{m0}$ tend to be underestimated by up to 1 m (see December, 20 event on Fig. 7). The discrepancies between modelled and observed $T_p$ are generally related to quick shifts under multi-modal sea states, i.e. from DSS ($T_p$ ranging from 10 s to 17 s) to local wind sea ($T_p$ ranging from 5 s to 8 s) or conversely.





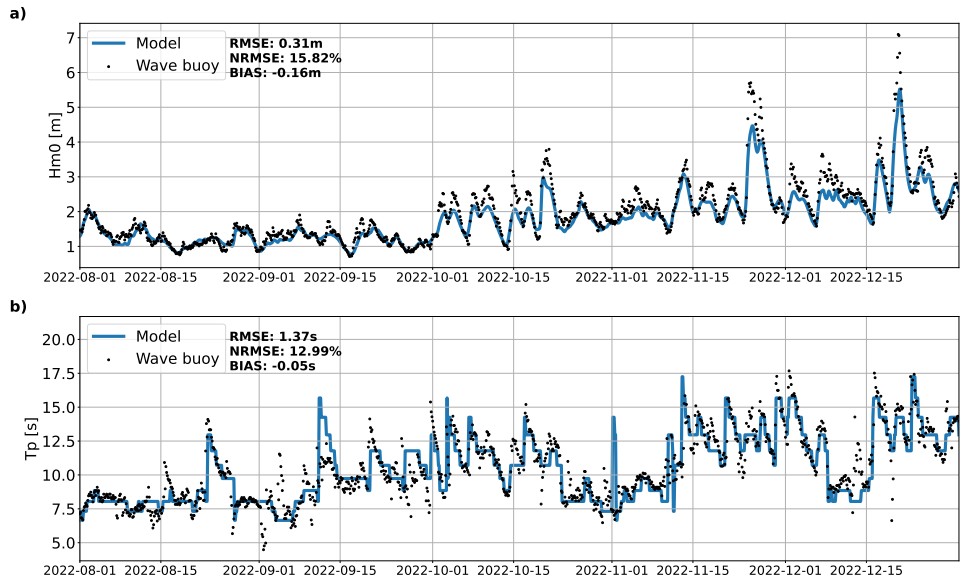

**Figure 7.** Wave bulk parameters estimated from wave buoy against model for August to December 2022 at Kauai (Hawaii) for: a) $H_{m0}$ and b) $T_p$.

### 3.2.3 Guadeloupe

Figure 8 depicts the comparison between model and wave buoy data at Pointe de la Grande Vigie to the North of Guadeloupe (90 m water depth) in the Caribbean Sea for $H_{m0}$, $T_{m02}$ and $T_p$. This figure reveals a very good behavior of the model with NRMSE under 12% for the wave bulk parameters. The strongest DSS event over the observed period (March to July, 2008) is correctly captured in terms of wave periods but the maximal $H_{m0}$ peak is again underestimated by up to 1 m. This problem will be further discussed in this paper.





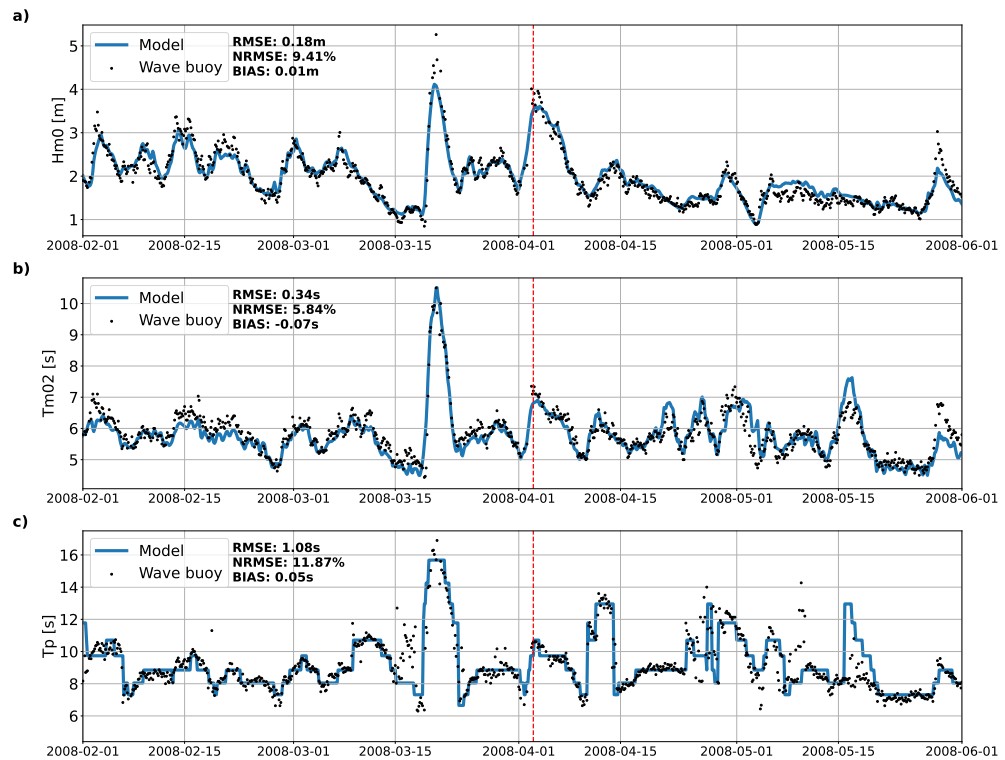

**Figure 8.** Wave bulk parameters derived from wave buoy against model for March to July 2008 at La Pointe de la Grande Vigie (Guadeloupe) for a) $H_{m0}$ b) $T_{m02}$ and c) $T_p$. The red line corresponds to a particular time where the PSD will be compared in the discussion.

### 215  3.2.4  New-Caledonia

Figure 9 displays the comparison between the model and the wave bulk computed from the pressure sensor data at Noumea (11 m water depth) for $H_{m0}$, $T_{m02}$ and $T_p$. The NRMSE and bias for $H_{m0}$ are 16.75% and 0.07 m, respectively. The periods are also correctly represented with NRMSE values of 7.92% and 14.09% for $T_{m02}$ and $T_p$, respectively. Similarly to Kauai, rapid fluctuations in multi-modal sea states are difficult to represent in the model and result in higher error on $T_p$.



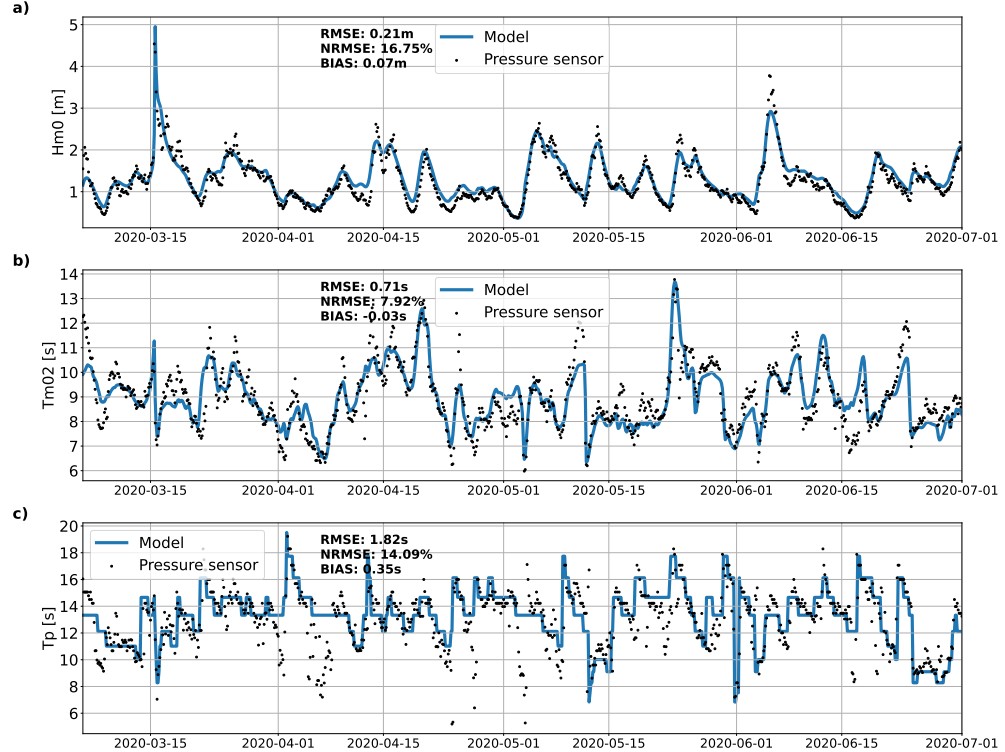

**Figure 9.** Wave bulk parameters estimated from pressure sensor against model for 2020 at Noumea (New-Caledonia) for: a) $H_{m0}$ b) $T_{m02}$ and c) $T_p$.

## 4 Discussion

### 4.1 Predictive skills and comparisons to existing global hindcasts

Focusing first on the sole wind correction by the quantile-quantile approach, better $H_{m0}$ predictions are obtained compared to the results obtained with the wind correction and source term of Alday et al. (2021). The approach presented in this study reduces the positive bias present for calmer conditions and consequently improves the general wave representation. It should be mentioned that the comparison was performed using the same wind correction and source term as T475 but the whole configuration of the model presented in Alday et al. (2021) also included a multigrid approach, improved ice forcing and ocean current forcing. In our configuration, one single structured grid is used with standard ice forcing and no current effect. One can expect better results combining the comprehensive configuration of Alday et al. (2021) with our wind correction.





Existing hindcasts such as ERA5 can result in poor predictive skills when compared with coastal measurements (Samou
et al., 2023). Similarly, coastal comparisons of the existing hindcasts ERA5-I and CFRS-W are limited by their global resolution
of 0.3°and 0.5°, respectively (Stopa and Cheung, 2014). Global unstructured grid models emerged recently (Brus et al., 2021;
Mentaschi et al., 2023), which avoid the use of the obstruction masks required for regular grids. However, the resolution
employed in these studies also remains too coarse to allow for a direct validation in shallow depth, that is very close to shore
due to the steep slopes usually surrounding tropical islands. Specifically designed to simulate sea state around tropical islands,
our new model is able to explicitly represent small islands and to reach a local resolution allowing a direct comparison with
nearshore observations and a correct representation of wave transformation from deep to shallow water. Even multigrid system
such as the ones described by Rascle and Ardhuin (2013) refined at 3' resolution or more recently Dutheil et al. (2020) and
Alday et al. (2021), both refined at 0.05°are too coarse to make a direct comparison with coastal observations, often located
only a few hundred meters from the coast.

**4.2    Remaining challenges in deep water**

However, despite these advances, areas such as polar regions, semi-enclosed seas, the Mozambique channel and the inner seas
of Indonesia displays larger errors compared to the other ocean basins (Fig. 4). In polar regions such as the Antarctic, overes-
timated $H_{m0}$ are likely related to inaccuracies in the representation of sea-ice dissipation, despite efforts to include icebergs
(improvement of the NRMSE of 0.1% when the icebergs were added in the present model). The sea ice concentration in ERA5
remains uncertain, especially in the Marginal Ice Zone (MIZ) (Renfrew et al., 2021). The oversimplified parameterisation of
wave propagation within sea ice areas does not take into account the calving and drifting of icebergs into the Southern Ocean,
which causes significant blocking of the wave energy (Ardhuin et al., 2011; Khan et al., 2021). Moreover, the altimetry data
used for model assessment can suffer inaccuracies in the presence of ice (Dodet et al., 2020).

Around coastlines and in semi-enclosed seas, model errors can result from wind inaccuracies associated with the land-sea
transition (Xie et al., 2001; Chelton et al., 2004). In addition, the ERA5 wind is known to be less accurate in mountainous areas
such as in the Mediterranean Sea, where the steep orography is misrepresented by the limited resolution of the reanalysis (Graf
et al., 2019; Dörenkämper et al., 2020; Gutiérrez et al., 2024). In the Mozambique Channel, the higher errors are probably due
to the fact that currents are not represented in our model. Indeed, the strong Agulhas Current has a direct influence on the wave
field propagation (Ardhuin et al., 2017). Similarly, in the Southern Ocean, accounting for the circumpolar current is known
to significantly reduce the positive bias of the modelled wave height (Rapizo et al., 2018). Although currents are known to
improve the model results (Marechal and Ardhuin, 2021), one of the main priorities was to produce a hindcast consistent over
time. Considering that CMEMS-Globcurrent surface currents are only available from 1993, it was decided to neglect current
in the model.

At the wave event scale, the $H_{m0}$ peaks tend to be underestimated at some locations for the most energetic events. In
order to investigate if this problem is already present in the deep ocean, Fig. 10 provides a comparison during the major 2007
DSS event between UG implicit and SG explicit results and data deduced from satellite altimetry. This comparison reveals
a diffusion behaviour of the UG run with an underestimation of the peak up to 0.7 m. Indeed, a possible explanation for the





underestimation of the peaks of $H_{m0}$ can be the diffusion of the implicit schemes used for the UG (Roland and Ardhuin, 2014; Abdolali et al., 2020).

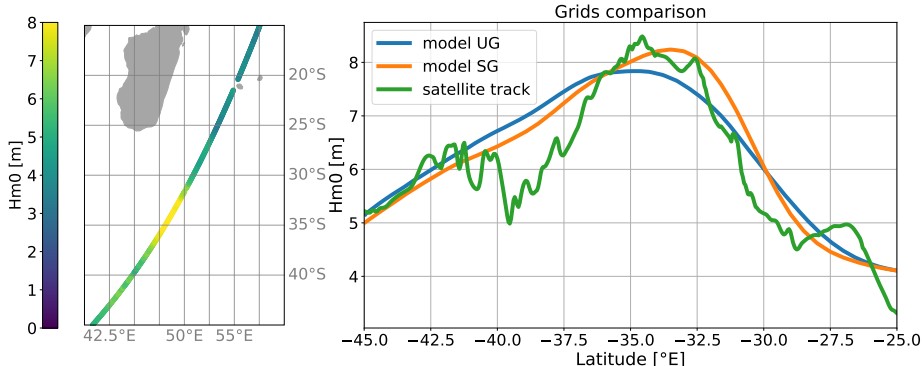

**Figure 10.** Comparisons between altimetry and modelled $H_{m0}$ WW3 numerical schemes comparison at 52E, 25S during the major DSS on May 13th, 2007.

So far, the latest version of WW3 with implicit schemes was only tested and validated for hurricanes (Abdolali et al., 2020, 2022) with short fetches and relatively small propagation area compared to the DSS example considered here. Further research efforts are therefore required to develop new higher order implicit schemes able to reduce diffusion and better capture $H_{m0}$ peaks. A second-order implicit scheme must remain positive and monotone, even at high CFL numbers, which is inevitable when increasing the resolution along the coasts. For example, the Crank-Nicholson time discretization maintains monotonicity only up to a CFL number of 2. Beyond this, achieving positivity and monotonicity becomes challenging because, according to the Godunov theorem, the scheme must be nonlinear.

Keeping in mind these limitations, the present approach based on implicit schemes remains an efficient compromise to simulate the generation and propagation of sea states down to nearshore regions, while maintaining low computational cost. Modelling the sea states at global scale with fine refinement at coastal scales is hardly practical with explicit schemes as simulating one year with 200 cores takes over 50 h. The simulation presented in this study was only possible thanks to the use of implicit schemes which overcomes the CFL constraint, and therefore considerably reduces the simulation time (Abdolali et al., 2020). Indeed, for the same computational resources, a one year simulation with implicit scheme took 12 h.

In order to further investigate the discrepancies observed, a comparison between modelled and observed power spectral density is shown at Pointe de la Grande Vigie (see Fig. 11).



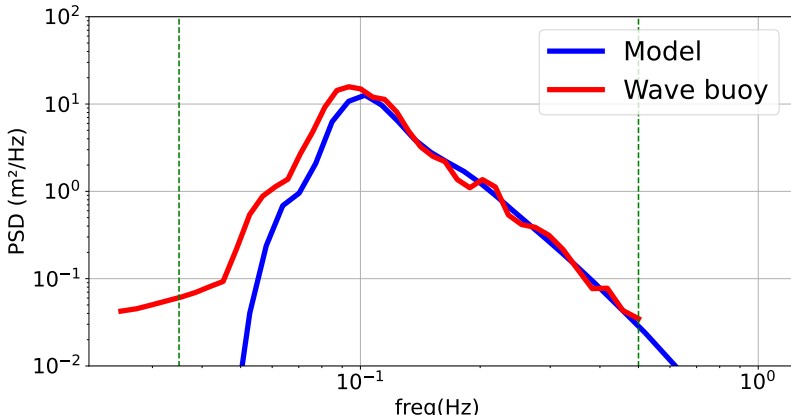

**Figure 11.** Comparison between power spectral density modelled and observed on at 18:00 UTC on April 2nd, 2008 (Pointe de la Grande Vigie, Guadeloupe).

This comparison shows an underestimation of the wave energy spectrum between 0.04 Hz and the peak frequency. This underestimation can be due to the parameterisation of the nonlinear quadruplet wave interactions, which is done with the Discrete Interaction Approximation (DIA) of Hasselmann et al. (1985). Although widely used, the DIA is a crude approximation as shown by Benoit (2007), Vledder et al. (2012) and Alday and Ardhuin (2023). Other alternative methods exist (Webb, 1978; Tracy and Resio, 1982; Hasselmann and Hasselmann, 1985; Masuda, 1980) but require very high computational costs and are therefore not practical with a global model. The underestimation of the wave energy spectrum below 0.04 Hz will be discussed in the next section.

### 4.3 Remaining challenges in coastal water

At global scale, the comparison between altimeter data and modelled $H_{m0}$ results in a NRMSE of 11.6%. In coastal zones, the comparison against *in-situ* data shows $H_{m0}$ NRMSE values between 9% and 22%, i.e generally consistent with the error observed in surrounding deep waters. Very close to shore and in shallow depths (La Reunion and New-Caledonia), the model also shows comparable predictive skills which demonstrates that the wave transformations from deep to shallow water are correctly simulated, despite challenges associated with the presence of steep-slope. Both modelled wave periods $T_{m02}$ and $T_p$ displayed similar accuracy than for $H_{m0}$ (NRMSE values<20%) although $T_p$ remains more unstable under multi-modal sea states.

Part of the discrepancies closest to shore can be attributed to the finest mesh resolution of 100 m (Fig. 12). Even though it is a very high refinement for a global model with a resolution of 50 km in deep waters, it remains too crude to represent the rapidly varying bathymetry related to the complex reef (Buckley et al., 2016). Gaining more insights on the adequate spatial resolution will require comparisons with measures and we are waiting for forthcoming high-resolution simulations to carry out comparative studies.





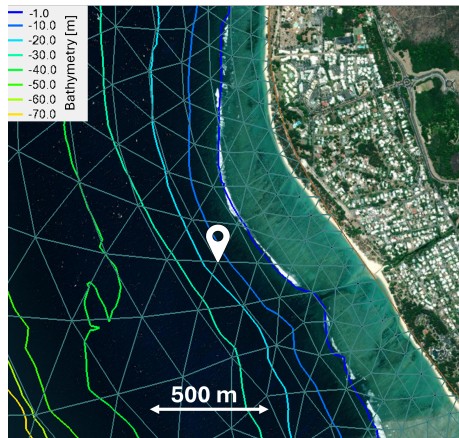

**Figure 12.** Satellite view of Hermitage fringing reef (La Reunion Island) with the extension of the model grid, a simplified bathymetric map and the location of the pressure sensor on the reef.

The steep bathymetries usually bordering tropical islands (with bottom slope locally easily exceeding 1:10) poses an additional challenge for spectral models. Indeed, spectral models are based on the wave action equation, which was derived for waves propagating in "slowly varying media" (Bretherton and Garrett, 1968), a condition that is not met for such steep slope.

When the waves approach the coast, triad interactions transfer energy from the peak frequency to lower (subharmonics) and higher (superharmonic) frequencies. In this model, the Lumped Triad Approximation (LTA) of Eldeberky (1996) is used to
reproduce the energy transfer towards higher frequencies. Although widely used, the LTA is a crude approximation and can only represents the second harmonic, which can result in high energy differences at high frequencies as seen in Fig. 13. The mean period $T_{m02}$ being sensitive to high frequencies, this problem could explain why we have higher errors on this parameter in shallow depths (as seen in Fig. 6).



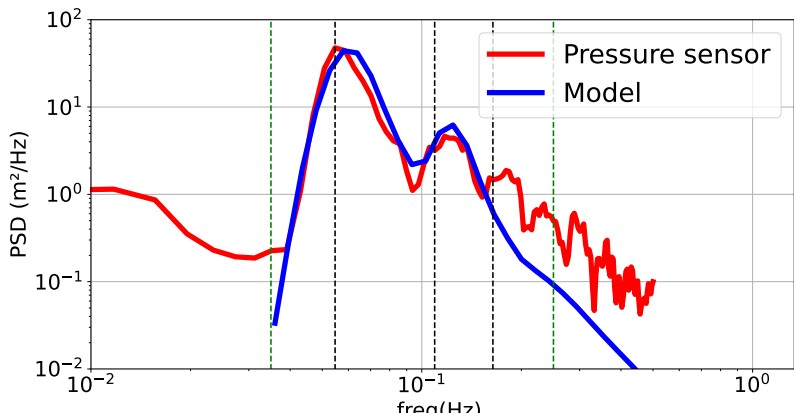

**Figure 13.** Comparison between measured power spectral density modelled and observed at 15:00 UTC on June 4th, 2022 (Hermitage, La Reunion Island). The green lines represent the cut-off frequencies. The black lines represent the peak frequency and the superharmonic frequencies.

## 5    Conclusions

A new global high resolution wave model for the tropical ocean is presented in this paper. The model aims at better simulating the sea state in the tropical ocean, which requires to capture the large extent of DSS propagation (i.e global model) while having a high local refinement to represent explicitly islands (i.e high refinement). Our approach combines the use of unstructured grid, with a global resolution of 50 km refined down to 100 m in nearshore areas of interest, and an implicit scheme which substantially reduces computational cost. In addition, we develop a new quantile-quantile correction for the ERA5 wind,

resulting in small biases for most wave heights. In addition to classic validation in deep waters against satellite altimetry data, this modelling strategy allows for the first time to perform direct comparisons in shallow depth, that is very close to shore due to the steep slopes usually surrounding tropical islands.

Despite difficulties related to the possible diffusion of the implicit scheme and the representation of nonlinear interactions, the model gives promising results with a correct representation of the wave transformation, from deep to shallow water. The

availability of long-term *in-situ* data remains one of the main challenges to validate wave models in tropical volcanic islands. The lack of wave measurements is usually due to the combination of remote location, difficult access, very steep seabed slopes and strong exposure to waves, which makes deployment in shallow areas even more challenging. The availability of spectrum data is be particularly helpful in understanding the origins of possible discrepancies between model and observations. In addition to the need to pursue the deployment of local *in-situ* surveys, the new satellite data, such as the recently launched

Surface Water and Ocean Topography (SWOT) mission, should provide useful high resolution wave measurements close to shore (Chelton et al., 2022) and facilitates validation of global and coastal models.



*Code and data availability.* Model setup files for the unstructured grid can be accessed at https://doi.org/10.5281/zenodo.13341123.

*Author contributions.* A. Gaffet: Conceptualization, Methodology, Software, Writing - original draft. X. Bertin: Conceptualization, Methodology, Software, Writing - review & editing, Supervision. D. Sous: Conceptualization, Writing - review & editing, Supervision. H. Michaud: Software, Writing - review & editing. E. Cordier: Data curation, Writing - review & editing . A. Roland: Software, Writing - review & editing.

*Competing interests.* The authors declare that they have no conflict of interest.

*Acknowledgements.* A. Gaffet is supported by a PhD fellowship from the CREOCEAN engineering consulting company. This research is also funded by the France 2030 PPR Ocean and Climate project FUTURISKS (ANR-22-POCE-0002). Permanent pressure data are collected on the reef slope at the Hermitage in the scope of the National Observation Service DYNALIT, part of the Research Infrastructure ILICO. We are very grateful to the CEREMA and the NDBC for sharing wave buoy data. We also thank the GEOCEAN-NC campaign for their pressure sensor data. The authors acknowledge the WW3 IFREMER/LOPS team for their satellite/model comparison Fortran routine. We are also thankful for the support of the Fondation de la Mer. This project was provided with computing HPC and storage resources by GENCI at IDRIS thanks to the grant 2023-AD010114543 on the supercomputer Jean Zay CSL partition.



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
