# Peer review of "A new global high resolution wave model for the tropical ocean using WAVEWATCH III version 7.14"

_EGUsphere, 2024_

## Author Comment (AC1)

**Reply on RC1**

We would like to thank the reviewer for his/her careful reading and constructive criticisms. The suggested modifications allowed us to improve our manuscript substantially.

This paper summarizes the results of a global unstructured mesh configuration for the WW3 model that uses high resolution to resolve small islands in the tropical ocean. The model spans resolutions between 50km and 100m, and the paper demonstrates its ability to capture the impacts of distant-source swells on tropical island costs. This global island-resolving configuration is novel and results presented are impressive. Overall, the message of the manuscript is clear and the text is well organized and well written. Below, I have some comments and suggestions that I hope will strengthen and help clarify the paper prior to publication.

General Comments:

1.  The resolution used around islands could be clarified a bit. The sentence starting in line 89 states: "Around the selected validation sites (see Sect.2.2.2), the mesh is further refined, reaching a resolution of 100m at La Reunion island where field observations are available close to shore." Calling out La Reunion here makes it unclear whether the 100m resolution extends to all validation sites, or just La Reunion. From Figure 1, it looks like the validation sites have different minimum resolutions, this should be more explicitly stated and explained. Table 1 may be a good place to summarize the minimum resolutions around each validation location. Also, the refinement criteria would be good to discuss. For example, how is the resolution function based on distance to coast, depth dependent, bathymetric slope dependent, etc.

We agree that the description of spatial resolution in our UG grid was not totally clear and we propose to reformulate this section in lines 91 to 94 as follows: "*Around the selected validation sites, the spatial resolution ranges from 1000 m to 500 m. At la Réunion Island, where a permanent pressure transducer is deployed 400 m from the shoreline only, the spatial resolution was further refined to 100 m. Such a fine resolution allows the water depth in the model to match that where the pressure transducer is located, which is essential to provide a consistent comparison*".

2.  The authors mention islands smaller than 10km² were removed, which makes sense from a total node-count perspective. However, these could still be parameterized with the source term dissipation approach from Mentaschi et al. 2018. Was this considered?

One of the main objectives of this study was to evaluate the relevance of explicitly representing islands in an UG compared to using obstruction grids. We believe that we provide a good demonstration of this modelling strategy, although using obstruction grids for very small Islands could potentially further improve our results, namely in areas with gazillions of small Islands such as in Indonesia. However, we think that this would fall outside the scope of the present study and prefer to leave this as a perspective in the discussion, which was added from lines 283 to 287 in the revised manuscript: "*Although the explicit representation of Islands whose areas are larger than 10 km² in our UG considerably improves wave predictions in archipelago*

*compared to obstruction grids, this threshold remains arbitrary. One can wonder if representing smaller islands could further improve the predictions, namely in areas with thousands of islands such as Indonesia. A possible alternative would be to combine the present approach with obstruction grids for the smallest islands, following, for instance, the method of Mentaschi et al. (2018)."*

3.    In Figure 1, it would be more helpful to show the mesh resolution color contours in the global image rather than the bathymetry. The bathymetry is more helpful in the zoom boxes where the actual mesh is overlaid. Also, the global NRMSE results show large improvements in the areas around the Solomon Islands and French Polynesia. Presumably these are 1km resolution regions. It may be helpful to have a figure showing the mesh in these areas to give a sense of the resolution provided in those regions, since it clearly has a large contribution to reducing local errors.

Indeed, the best would be to have vectorial figures for both the bathymetry and the mesh resolution of the grid but such figures are too heavy to meet the journal policy so we added separate figures for the bathymetry and the mesh resolution in the revised version.

4.    I think the bias correction approach could be clarified starting in Line 108. Is the correction applied as a piecewise multiplication factor? Is this a correct way to express the correction: W = W for W<15m/s, W=1.1*W for 20m/s<W<25m/s, W = 1.15*W for W > 25m/s?. It may also be good to show a figure for the bias corrected winds similar to Figure 2.

Indeed, the proposed correction is based on a piecewise multiplication factor. As we expect this correction to be useful for the scientific community, we propose to present it in the form of a table and add the following description in Line 116: "*The proposed correction is based on a piecewise multiplication factor displayed in Table 1.*"

| Wind speed [m/s] | Correction factor [-] |
|---|---|
| <17 m/s | 1 |
| 17 m/s < $U_{10}$ < 20 m/s | 1.03 |
| 20 m/s < $U_{10}$ < 23 m/s | 1.08 |
| 23 m/s < $U_{10}$ < 26 m/s | 1.14 |
| 26 m/s < $U_{10}$ < 30 m/s | 1.20 |
| $U_{10}$ > 20 m/s | 1.27 |

5.    Since the ERA5 bias correction strengthens the high winds, it seems counterintuitive that a positive bias is present for the uncorrected ERA5 SG model, while the ERA5 correction results in a negative bias for the SG model. Perhaps I am misinterpreting the bais values, but I'm used to thinking of a positive bias meaning the model is overpredicting observations (and a negative indicating it is underpredicting) Some explanation for this would be helpful.

Sure, in order to make that point clearer, we added in the revised paper in lines 198 to 201: "*Indeed, the T475 parameterisation uses a higher Betamax parameter, which controls the wind input term and hence the wave growth, set to 1.75 compared to 1.43 in the ERA5_QC parameterisation. This higher Betamax leads to greater wave growth and therefore a positive bias, although winds are lighter in the T475 than in the ERA5_QC.*"

6.    In Figure 5, it is interesting that overall, the uncorrected SG has a lower absolute bias than either of the ERA5_QC models for the 10-12m wave bins. I would expect the ERA5 bias correction to have the greatest effect here.

Here we agree but we like to mention that such large waves are driven by extreme winds (i.e. >25-30 m/s), for which we have a limited number of observations, as it can be seen on figure 2. We therefore recognize that our wind correction could be further improved for very high winds, provided that more observations are available, which is now discussed in Section 4.1 from lines 240 to 244:

"*However, as seen in Fig 6, the uncorrected SG has a lower absolute bias than either of the ERA5_QC for the 10-12 m wave bins. Such large waves are driven by extreme winds (i.e. larger than 25-30 m/s), for which we have a limited number of observations, as can be seen on Fig. 2. The quantile-quantile wind correction could be further improved for very high winds, provided more observations become available.*"

7.    The results in Section 3.2 are quite good, but I feel like they could be further strengthened and made more impactful by showing how the wave bulk parameters improve with resolution. For example, it would be really useful to see what 100m resolution buys you in terms of accuracy vs the 1km used for other islands. This would give the reader more guidance on the right level to use for their given application. I understand there may be restrictions here as it relates to resolving the actual observation location. I also understand this would mean developing a new mesh and running more simulations, which may not be practical. However, some commentary on whether 100m there is likely to be a difference between, say 100m and 200m resolution, would be helpful. This is another reason it would be good to be more clear about the local resolution used (see first comment) because that could provide these types of insights as well.

We modified our grid in order to degrade the spatial resolution to 1000 m at La Réunion (see Figure below for the comparison between the two meshes). However, considering the steep bathymetric slopes surrounding the Island, the refinement was too coarse to allow for a consistent water depth in the model and at the location of the pressure sensor so we had to move the output point. Therefore, it is difficult to determine whether the poorer results with the 1000 m resolution are due the poor representation of fast-varying water depths or if they are caused by an interpolation issue. In order to make that point clearer, we added in the revised version of the paper in lines 329 to 332 : "*In order to make clear recommendations concerning the minimum resolution required for such comparisons, we need more adequate measurements and we are waiting for forthcoming high-resolution simulations to carry out comparative studies. Additionally, the issue of resolution remains closely tied to interpolation issues when comparing measurements and model in rapidly varying media.*"

[Figure]

Pressure sensor Reeftemps/DYNALIT and Model Comparison - Hermitage refined at 100m and 1km

*Figure 1 : Wave bulk parameters derived from the pressure sensor against the model for grid refinement up to 100 m and 1 km, respectively, for April to August 2022 at Hermitage (La Reunion) for a) Hm0 b) Tm02 and c) Tp. The blue and orange lines show the results obtained with a grid refined to 1km and 100m, respectively.*

8.     More could be said about the computational performance of the model. Line 277 states: "Indeed, for the same computational resources, a one year simulation with implicit scheme took 12 h." What was the number of MPI ranks used for this to get this throughput and how does the parallel scalability of this configuration compare to Abdolali et al., 2020)?

200 MPI ranks were used in the simulation described, which corresponds to 10 nodes with 20 cores per node. The very good model performance and scalability, as Abodali et al. 2020 showed in their paper, are shown one figure 2:

[Figure]

*Figure 2 : Model performance (v7p14) on HPC environment and Scalability of WW3 model for implicit numerical solver for an unstructured grid with 290k nodes with ~100m minimum resolution. The horizontal axis shows the number of computational cores and the vertical axis represents the performance of the model.*

Specific Comments:

1. The sentence starting in line 35 should better distinguish between the obstruction mask approach in Tolman 2003 and the approach in Mentaschi et al. 2018. The latter of these is based on a dissipation source term can be for both structured and unstructured grid models.

We thank you for pointing this problem out, we have now modified this part in the revised version and provided additional elements to better distinguish the two approaches in lines 36 to 41: "*To simulate the effect of small islands on the wave field while keeping computational times acceptable on a global scale, two approaches were developed. The first one uses an obstruction mask technique, which is only applicable to structured grids. In this approach, the percentage of land within each cell is used as a coefficient to calculate the attenuation of the wave action flux through the considered cell (Tolman, 2003). The second approach is based on a source term that considers both the attenuation of the wave action flux through the considered cell and its shadowing effect on the downstream cells. This method can be used for structured and unstructured grids (Mentaschi et al., 2018)*"

2. The latest release version of WW3 on the NOAA-EMC repo is 6.07.1, so more information should be provided about what the version 7.14 used in this study represents (and how to access it).

You are right, information about the version 7.14 is now added in the revised version of the paper in the "Code and availability" section: "*The version 7.14 of WAVEWATCHIII is available here https://github.com/NOAA-EMC/WW3. Specifically, the code used in this work can be found at 10.5281/zenodo.14011562*"

3. In the paragraph starting with line 65, triad interactions could be listed as an important source term for shallow water along with bottom friction and breaking-induced dissipation.

You are right, we kept nonlinear interactions, triad and quadruplet interactions together but it makes more sense to move the triad in the shallow water section along with bottom friction and breaking-induced dissipation. It is now done in the revised version.

4.      I would recommend adding the dates used to validate each site to Table 1.

Combining the recommendation you made in the first comment and this one, we modified Table 1 with the minimum resolutions around each validation location and also the dates used to validate each site in the revised version in line 75.

5.      The text in Section 3.1 was a little unclear to me at first reading (although certainly made sense when I read it more carefully). I would recommend a slight tweak to make this a little more clear: "Two different comparisons are performed in this section. The model validation in deep water is first performed *only* on structured grids to evaluate the effect of wind field correction. *Next,* the evaluation of the spatial discretisation is carried out comparing SG and UG approaches, both including the wind correction."

We thank the reviewer for his/her proposition of modification which is included in the revised version of the paper in lines 176 to 178.

6.      Even though the NRMSE and bias metrics are standard, I think they should be explicitly defined in the paper for completeness.

We agree with you and have added a subsection in the Material and Methods called "Validation metrics" in lines 168 to 173 where we introduce the NRMSE and bias equations for completeness.

7.      The overall mean NRMSE and NBIAS values in red on the plots in Figures 3 and 4 are difficult to read, larger text in black would be better.

We thank the reviewer for its attention to details. Following your advice, the NRMSE and NBIAS values are now easier to read in the revised version.

8.      I would recommend that the paragraph starting in Line 278 be moved ahead of the discussion of the two previous paragraphs on performance and numerics. I think this would flow better since now it shifts somewhat abruptly back to discussion of model results vs observations.

That is a good point and is taken into account in the revised version.

9.      Line 285 that "The underestimation of the wave energy spectrum below 0.04 Hz will be discussed in the next section." I didn't see this discussion there. The next section seems to focus on the high frequency underestimation (Figure 13).

You are right, this part of the discussion was missing, we have added in lines 329 to 332: "*In coastal zones, the underestimation of the energy spectrum below 0.04 Hz that can be seen on Fig.14 and 12 can be attributed to the presence of infragravity waves (see Bertin et al. (2018) for a recent review), which are not represented in the present model. At Grande Vigie station*

*(Fig. 12), the 90 m water depth implies that energy in the IG wave band corresponds to free IG waves reflected along the coast, a problem discussed by Alday et al. (2021).*"

---

## Author Comment (AC2)

**Reply on RC2**

We would like to thank the reviewer for his/her careful reading and constructive criticism. The suggested modifications have been a decisive help to revise and improve our manuscript.

This paper presents a global unstructured mesh configuration of the WW3 model tailored for capturing distant-source swells in the tropical ocean, especially around small volcanic islands. The authors implemented the spectral wave model WAVEWATCH III over a variable-resolution grid, with finer mesh around island shorelines (as low as 100m), allowing for a more detailed representation of tropical islands. The model, forced by ERA5 wind fields and corrected for biases via satellite data, seeks to address limitations in previous models that relied on coarse, regular grids with obstruction masks, which introduced negative biases. The new approach demonstrates improved predictive accuracy for sea states in tropical areas, and results are compared with in-situ data nearshore at depths between 10-30m. However, several aspects need refinement, such as clarifying resolution differences around validation sites, examining triad interactions and partition comparison in shallow water, and addressing wind field temporal resolution. The manuscript's clear and well-organized structure contributes valuable insights into the challenges and methodologies for wave modeling in complex island environments.

**Major Comments:**

1. **Clarification of Novelty:** The claim that this is the first study to directly compare with nearshore data in 10-30m depths is inaccurate; other global wave model studies have also achieved this. While this study is pioneering in certain aspects, the authors should revise the abstract and relevant sections to reflect this context accurately.

Here we only partly agree with this comment because the very few global wave model studies that made direct comparisons with nearshore data considered wave buoys actually moored by 20-30 m water depth but located several km away from the coast (Zheng et al., 2016; Alday et al., 2021). The novelty of our study is that, due to the volcanic island context, such water depth are found a few hundred meters from shore only. In the revised manuscript, we better explained the novelty of our study in abstract and in the discussion.

We agree with the reviewer's remark, the following modifications have been applied in the revised version of the manuscript in lines:

In lines 11 to 13 in the Abstract:

*"Moreover, this new simulation allows for the first time direct comparisons with the in-situ data collected on volcanic islands at depths of water ranging from 10m to 30m, which corresponds to a few hundred meters from shore."*

In lines 257 to 262 in the Discussion:

*"However, the resolution employed in these studies also remains too coarse to allow for a direct validation in nearshore shallow depth. A few studies already compared global wave model with stations located by 10-30 m water depth, although these wave buoys were moored at gently sloping inner shelves or in big lakes (Zheng et al., 2016; Alday et al., 2021). The novelty here*

*is that, due to the steep slopes usually surrounding volcanic islands where our stations are located, such water depths are found very close to shore, typically a few hundred meters.''*

2. **Wind Field Temporal Resolution:** The choice of 3-hourly ERA5 data might be insufficient for fast-moving systems, such as hurricanes. Given that ERA5 offers hourly data, it would be valuable to understand why 3-hourly data was chosen. Further, suggestions on time interpolation techniques to better capture these conditions would be helpful.

Hourly wind field data were also considered but a sensitivity analysis revealed that it only improved model bias predictive skills by 0.62% and increased the normalized error by 0.04% (see Figure 1). In order to explain our choice we added in the revised version in lines 107 to 108 : ". *ERA5 also offers hourly data but sensitivity tests revealed similar results with 3 hourly wind fields.*"

[Figure]

*Figure 1 : Normalized RMS error (A) and bias (C) between Hm0 deduced from altimetry and simulated with the SG 1h-ERA5_QC configuration and normalized RMS error (B) and normalized bias (D) with the SG 3h-ERA5_QC configuration*

Concerning hurricanes, we added complementary explanations in lines 275 to 276: "*Moreover, the spatial resolution and the representation of the strongest winds limit our ability to accurately model waves generated by hurricanes (Jullien et al., 2024).*"

3. **Triad Interaction in Shallow Waters:** While the study employs triad interactions, the impact on shallow water gauges isn't clearly demonstrated. It would be beneficial to include an analysis of triad effects on gauge results in these areas.

We propose to show the wave spectrum in La Réunion Island at the peak of the DDS event of June 2022, with and without triad interactions (see Figure 2). In the revised version of the manuscript, Fig. 14 was modified accordingly:

[Figure]

*Figure 2 : : Comparison between measured power spectral density modelled and observed at 9:00 UTC on June 8th, 2022 (Hermitage, La Reunion Island). The green lines represent the cut-off frequencies. The black lines represent the peak frequency and the superharmonic frequencies.*

4. **Handling of Singularities with Regular Grid:** If the regular grid configuration masked out the North Pole to avoid singularities, clarify at which latitude this masking was applied and what is the consequences on the results?

The regular grid extends up to 80.5° latitude, excluding the North Pole. Beyond 80° latitude, the ocean is predominantly covered by ice, so extending the grid to higher latitudes would not provide additional meaningful information.

5. **Quantifying Swells without Partitioning:** The paper does not describe a method for separating wind-sea and swell partitions, which is essential for accurately quantifying swells (this is the main argument of this paper). Swell partitioning would allow better comparison between modeled and observed swells and enhance understanding of the model's performance across different wave types. This would be beneficial, especially in coastal observations, to assess model accuracy across partitions.

In order to separate wind-sea and swell partitions, directional spectrum data exposed to wind sea and swell are required. As the sensors deployed in La Reunion and New Caledonia are located along the leeward coast of these islands, they were not exposed to wind waves generated by trade winds and hence were not considered for this analysis. We focused on the Guadeloupe wave buoy, which was exposed to trade wind sea and for which directional spectra were available. Because wind measurements required to compute the wave age were not available at this station, the spectral partitioning using a frequency cut (PTM5 in WW3) seems the most adapted in our case. Considering frequencies where both wind waves related to trade winds and DSS are present, a minimum can be observed at about 0.12 Hz (see figure 3), which we used to separate swells and wind waves. Integrating the wave spectrum with this frequency cutoff, one can see that the model reproduces DSS and wind waves with a similar accuracy (see figure 4). However, as satellite data from the missions used in this study and most wave buoys do not

allow access to wave spectra, we do not think that this analysis should be integrated in manuscript.

[Figure]

*Figure 3 : Comparison between power spectral density modelled and observed at 21:00 UTC on March 22nd, 2008 (Pointe de la Grande Vigie, Guadeloupe). The purple line represents the frequency cut-off for the swell partitioning.*

[Figure]

*Figure 4 : Wave bulk parameters derived from partitioned wave buoy data against model for March to July 2008 at La Pointe de la Grande Vigie (Guadeloupe) for a) Hm0 b) Tm02 and c) Tp.*

**Minor Comments:**

1. **Figure 5 Labeling Issue:** Ensure the y-axis label in Figure 5 is fully visible.

We thank the reviewer for pointing that out, we have modified the figure in the revised version of the paper so the y-axis label is now fully visible.